# Switchable synthesis of natural-product-like lawsones and indenopyrazoles through regioselective ring-expansion of indantrione

Bingwei Hu[1,3], Wenxin Yan[2,3], Peiyun Jiang[1,3], Ling Jiang[1], Xu Yuan[1], Jun Lin[1], Yinchun Jiao[2 ✉] & Yi Jin[1 ✉]

Lawsones and indenopyrazoles are the prevalent structural motifs and building blocks in pharmaceuticals and bioactive molecules, but their synthesis has always remained challenging as no comprehensive protocol has been outlined to date. Herein, a metal-free, ring-expansion reaction of indantrione with diazomethanes, generated in situ from the N-tosyl-hydrazones, has been developed for the synthesis of lawsone and indenopyrazole derivatives in acetonitrile and alcohol solvents, respectively. It provides these valuable lawsone and pyrazole skeletons in good yields and high levels of diastereoselectivity from simple and readily available starting materials. DFT calculations were used to explore the mechanism in different solutions. The synthetic application example also showed the prospects of this method for the preparation of valuable compounds.

[1] Key Laboratory of Medicinal for Natural Resource, Ministry of Education and Yunnan Province, School of Pharmacy, Yunnan University, 650091 Kunming, China. [2] School of Chemistry and Chemical Engineering, Key Laboratory of Theoretical Organic Chemistry and Functional Molecular, Ministry of Education, Hunan University of Science and Technology, 411201 Xiangtan, China. [3]These authors contributed equally: Bingwei Hu, Wenxin Yan, Peiyun Jiang. ✉email: yinchunjiao@hnust.edu.cn; jinyi@ynu.edu.cn

The generation of privileged natural product scaffolds is a successful strategy for obtaining libraries of bioactive compounds[1–4]. Scaffold generation focuses on the application of methodologies capable of generating structures that cover a larger part of the chemical and biological space[5–7]. Since scaffold generation is pivotal during the early stage of drug discovery, issues like chemical efficiency and easy diversification are crucial in chemical reaction development. Using diazo compounds to expand the size of cyclocarbonyl compounds by one carbon unit is a classical homologation method for the generation of structurally complex and diverse scaffolds[8–11]. In the last century, the ring expansion of isatins and other cyclic ketones with multiple diazo variants (e.g., diazomethane and α-diazoesters) was reported[12–15]. Several catalytic ring-expansion reactions also have been performed by the Kingsbury[16], Maruoka[17], and Feng groups[18]. However, the ring-expansion reaction of benzo-fused cyclic dione substrate faces regioselectivity because of 1,2-aryl migration and 1,2-carbonyl migration reaction pathways (Fig. 1a). To the best of our knowledge, the regioselective ring-expansion reaction of indantrione has not been reported. In a continuation of our research with diazo compounds, we report herein the regioselective ring expansion of indantrione with α-aryldiazomethane involving 1,2-carbonyl migration (Fig. 2b).

Lawsones and indenopyrazole derivatives constitute important classes of polycyclic systems embedded in a plethora of natural products and synthetic compounds showing antimalarial (Atovaquone)[19], antipneumocystic (Buparvaquone)[20] bioactivities, and protease inhibitory activities such as cyclin-dependent kinase (CDK)[21,22], hypoxia-inducible factor (HIF)-1[23,24], and tubulin[25–27] (Fig. 2). In addition, lawsones are also the essential synthons[28] for the synergistic nucleophilicity and electrophilicity to overcome various of synthetic challenges[29–31]. Despite being appealing structural motifs, highly efficient approaches to prepare

lawsones and indenopyrazoles are limited[32–41]. The reported protocols are generally multi-step reactions, which require high loadings of metal catalysts and properly functionalized substrates that are not readily available. The solvent-controlled ring-expansion reaction of indantrione and α-aryldiazomethane would be a straightforward method for preparing these lawsones and indenopyrazole compounds. Herein, a metal-free, ring-expansion reaction of indantrione with diazomethanes, generated in situ from the N-tosylhydrazones, has been developed for the synthesis of lawsone and indenopyrazole derivatives in acetonitrile and alcohol solvents, respectively.

## Results and discussion

**Optimization of the reaction conditions**. We screened an array of solvents and bases (Table 1) for their reactivity and chemoselectivity in the ring-expansion reaction of indantrione **1a**, benzaldehyde **2a**, and *p*-methylbenzenesulfonohydrazide **3**. This survey led to the following optimized reaction conditions: (1) 1,2,3-Indantrione **1a** (1.0 mmol) and $Cs_2CO_3$ (2.0 equiv.) were added to α-aryldiazomethane and stirred at 80 °C in MeCN for 3 h. The desired lawsone **4a** product was obtained in 90% yield (Table 1, entry 1); (2) 1,2,3-Indantrione **1a** (1.0 mmol) and $Cs_2CO_3$ (3.0 equiv.) were added to the α-aryldiazomethane and stirred at 80 °C in EtOH for 10 h. The desired indenopyrazolone **5a** product was obtained in 83% yield with a dr value >95:5 (Table 1, entry 5). A small amount of lawson product **4a** was generated in a 15% yield. This result shows that solvent-controlled regioselective ring-expansion reaction had a 1,2-carbonyl migratory tendency with two different products.

**Substrate scope of indantriones and aldehydes for producing lawsones**. We then explored the scope of this reaction with a range of indantriones **1** and aldehydes **2** (Fig. 3) (see Supplementary Notes 1, 2) for producing lawsones. As anticipated, the ring-expansion reaction of indantriones generally tolerated a broad range of substituted aryl aldehydes bearing either electron-donating or electron-withdrawing substituents, such as methyl, fluoro, chloro, bromo, nitro, nitrile, trifluoromethyl, and naphthyl groups. The corresponding hydroxynaphtho-quinones were afforded in high to excellent yield. The electron-donating groups slightly reduced the reaction yield compared with electron-withdrawing groups (e.g., **4b**, *p*-F, 96% and **4j**, *p*-CF₃, 93% yields vs. **4e**, *p*-CH₃, 90% and **4h**, *p*-OMe, 85% yields). The position of substituents did not significantly affect the reaction yield (e.g., **4b**, *para*-F; **4m**, *meta*-F; **4u**, *ortho*-F; with yields of 96%, 95%, and 90%, respectively). Polysubstituted aryl aldehydes were also suitable for this reaction (e.g., **4aa** and **4ab** with 85% and 84% yields, respectively). Borate ester-substituted aryl aldehydes (**4t**, 91% yield), heteroaryl aldehydes (**4ad** and **4ag**, 93% and 81% yields,

**Fig. 1 Regiochemistry of the ring expansion of cyclic ketones. a** The selective ring expansion of isatins. **b** This work, the selective ring-expansion of indantrione.

**Fig. 2 Representative bioactive compounds.** Such as Lawsone derivatives with tubulin, CDK, and HIF-1 inhibitions and indenopyrazole derivatives with antimalarial and antipneumocystic activities.

**Table 1 Reaction optimization[a,b,c].**

| Entry | Solvent | Base | Time (h) | Yield (%) | |
|---|---|---|---|---|---|
| | | | | **4a** | **5(dr)** |
| 1 | CH₃CN | Cs₂CO₃ | 10 | 90 | nd |
| 2 | 1,4-dioxane | Cs₂CO₃ | 10 | 31 | nd |
| 3 | toluene | Cs₂CO₃ | 10 | nd | nd |
| 4 | THF | Cs₂CO₃ | 10 | 47 | nd |
| 5 | EtOH | Cs₂CO₃ | 10 | 15 | 83 (>95:5) |
| 6 | MeOH | Cs₂CO₃ | 10 | 8 | 87 (>95:5) |
| 7 | n-propanol | Cs₂CO₃ | 10 | 27 | 65 (>95:5) |
| 8 | H₂O | Cs₂CO₃ | 10 | 5 | nd |
| 9 | EtOAc | Cs₂CO₃ | 10 | 67 | nd |
| 10 | DMSO | Cs₂CO₃ | 10 | 51 | nd |
| 11 | CH₃CN | DBU | 3 | 85 | nd |
| 12 | CH₃CN | K₂CO₃ | 3 | 83 | nd |
| 13 | CH₃CN | n-BuOK | 3 | 15 | nd |
| 14 | CH₃CN | Na₂CO₃ | 3 | 77 | nd |
| 15 | CH₃CN | Et₃N | 3 | nd | nd |
| 16 | CH₃CN | NaOH | 3 | 35 | nd |
| 17 | EtOH | DBU | 10 | 3 | 80 (>95:5) |
| 18 | EtOH | K₂CO₃ | 10 | 5 | 74 (>95:5) |
| 19 | EtOH | t-BuOK | 10 | 35 | 62 (>95:5) |
| 20 | EtOH | Na₂CO₃ | 10 | nd | 77 (>95:5) |
| 21 | EtOH | Et₃N | 10 | nd | 36 (>95:5) |
| 22 | EtOH | NaOH | 10 | 46 | 51 (>95:5) |

nd not detected.
[a]Reagents and conditions: (1) In a 25 mL reaction tube, benzaldehyde **2a** (1.2 mmol), p-toluenesulfonyl hydrazide **3** (1.2 mmol), solvent (10 mL), under ambient atmosphere (1 atm), at room temperature for 0.5–1 h. (2) indantrione **1a** (1.0 mmol), base (2.0 mmol), at 80 °C for 10 h.
[b]The dr value was determined by ¹H-NMR analysis of crude products.
[c]Isolated yield based on **1a**.

respectively), and some large sterically hindered aryl aldehydes (**4af** and **4ah**, 85% and 80% yields, respectively) also reacted smoothly. The ring-expansion reaction also produced equally satisfying results using alkyl aldehydes, e.g., prenylaldehyde (**4ai**, 74% yield), butyraldehyde (**4aj**, 71% yield), and 2-ethylcaproaldehyde (**4ak**, 50% yield) as substrates. The corresponding ring-expansion products were also obtained under the standard conditions for substituted indantriones and benzaldehyde (**4al** 88%, and **4am** 85% yields). To verify the relative configurations of the hydroxynaphtho-quinones, **4h**, and **4ae** were selected as representative compounds and characterized by X-ray crystallography (see Supplementary Data 1, 2, and see Supplementary Note 3, Figs. S1 and S2) (CCDC 2149716 (4h), 2149715 (4ae), 2149717 (5a), 2149718 (5f), 2149719 (5q), 2149720 (5r) contain the supplementary crystallographic data for this paper. These data can be obtained free of charge from the Cambridge Crystallographic Data Centre via https://www.ccdc. cam.ac.uk/data_request/cif).

**Substrate scope of indantriones and aldehydes for producing indenopyrazoles.** Similarly, we investigated the scope of this reaction using a range of indantriones **1** and aldehydes **2** (Fig. 4) (see Supplementary Notes 1, 2) to produce indenopyrazolones. Under the standard reaction conditions, aryl aldehydes with various substituents, such as methyl, fluoro, chloro, bromo, or methoxy groups, smoothly reacted to afford the indenopyrazolone derivatives in high to excellent yields and high

stereoselectivities (dr, ≥95:5). Electron-donating groups also slightly reduced the reaction yield compared with electron-withdrawing groups (e.g., **5b**, p-CH₃, 83% and **5f**, p-OCH₃, 79% yield vs. **5c**, p-F, 91% and **5d**, p-Cl, 88% yields). The position and number of substituents did not significantly affect the reaction yield (e.g., **5c**, para-F; **5i**, meta-F; **5 m** ortho-F; with yields of 91%, 90%, and 81%, respectively). The reaction also produced equally satisfying results using heteroaryl aldehydes (e.g., **5q**, 85% yield) and sterically bulky aryl aldehydes (e.g., **5t**, 81% yield). In different alcohol solutions, the corresponding esterified indenopyrazolone derivatives were obtained, such as **5o–5ab**, with medium to excellent yields and high stereoselectivities (dr, ≥95:5). The corresponding products were also obtained under standard conditions for substituted indantriones and benzaldehyde (**5ad** 85%, and **5ae** 87% yields). Unfortunately, alkyl-substituted aldehydes were found to be incompatible with this transformation and did not yield any product. **5a**, **5f**, **5q**, and **5r** were characterized by X-ray crystallography (see Supplementary Data 3–6, and see Supplementary Note 3, Figs. S3–S6) (CCDC 2149716 (4h), 2149715 (4ae), 2149717 (5a), 2149718 (5f), 2149719 (5q), 2149720 (5r) contain the supplementary crystallographic data for this paper. These data can be obtained free of charge from the Cambridge Crystallographic Data Centre via https://www.ccdc. cam.ac.uk/data_request/cif).

**Mechanistic studies.** To elucidate the reaction mechanism of the regioselective ring-expansion of indantrione and α-aryldiazomethanes,

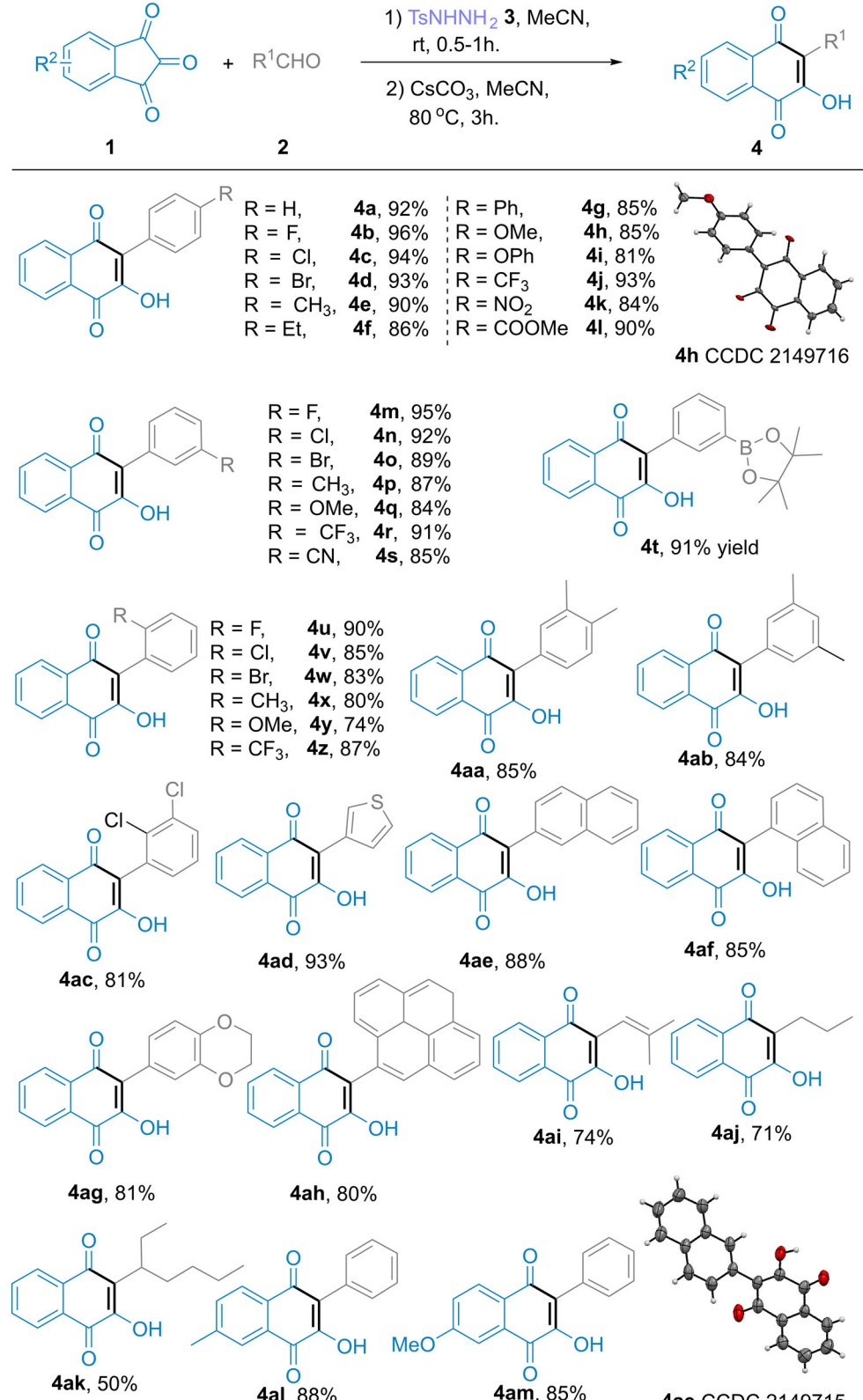

**Fig. 3 Substrate scope of indantriones and aldehydes for producing lawsones. a** General conditions: (1) In a 25 mL reaction tube, aldehyde **2** (1.2 mmol), *p*-toluenesulfonyl hydrazide **3** (1.2 mmol), CH$_3$CN solvent (10 mL), under ambient atmosphere (1 atm), stirred for 0.5–1 h. (2) indantrione **1a** (1.0 mmol), Cs$_2$CO$_3$ (2.0 mmol), at 80 °C for 3 h. **b** Isolated yield based on **1a**.

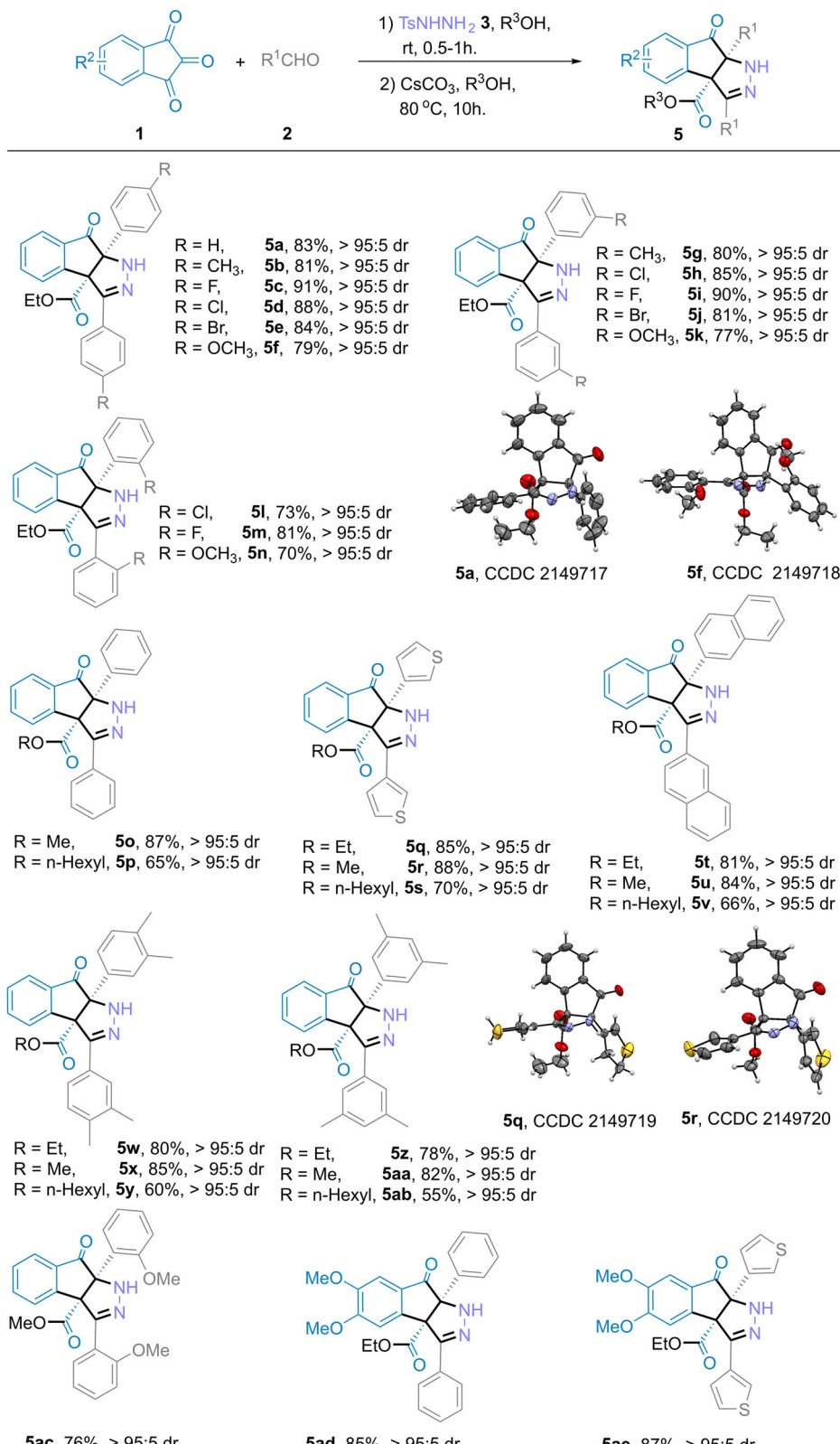

**Fig. 4 Substrate scope of indantriones and aldehydes for producing indenopyrazoles. a** General conditions: (1) In a 25 mL reaction tube, aldehyde **2** (2.1 mmol), p-toluenesulfonyl hydrazide **3** (2.1 mmol), R'OH solvent (20 mL), under ambient atmosphere (1 atm), stirred for 0.5–1 h. (2) indantrione **1a** (1.0 mmol), Cs₂CO₃ (3.0 mmol), at 80 °C for 10 h. **b** The dr value was determined by ¹H-NMR analysis of crude products. **c** Isolated yield based on **1a**.

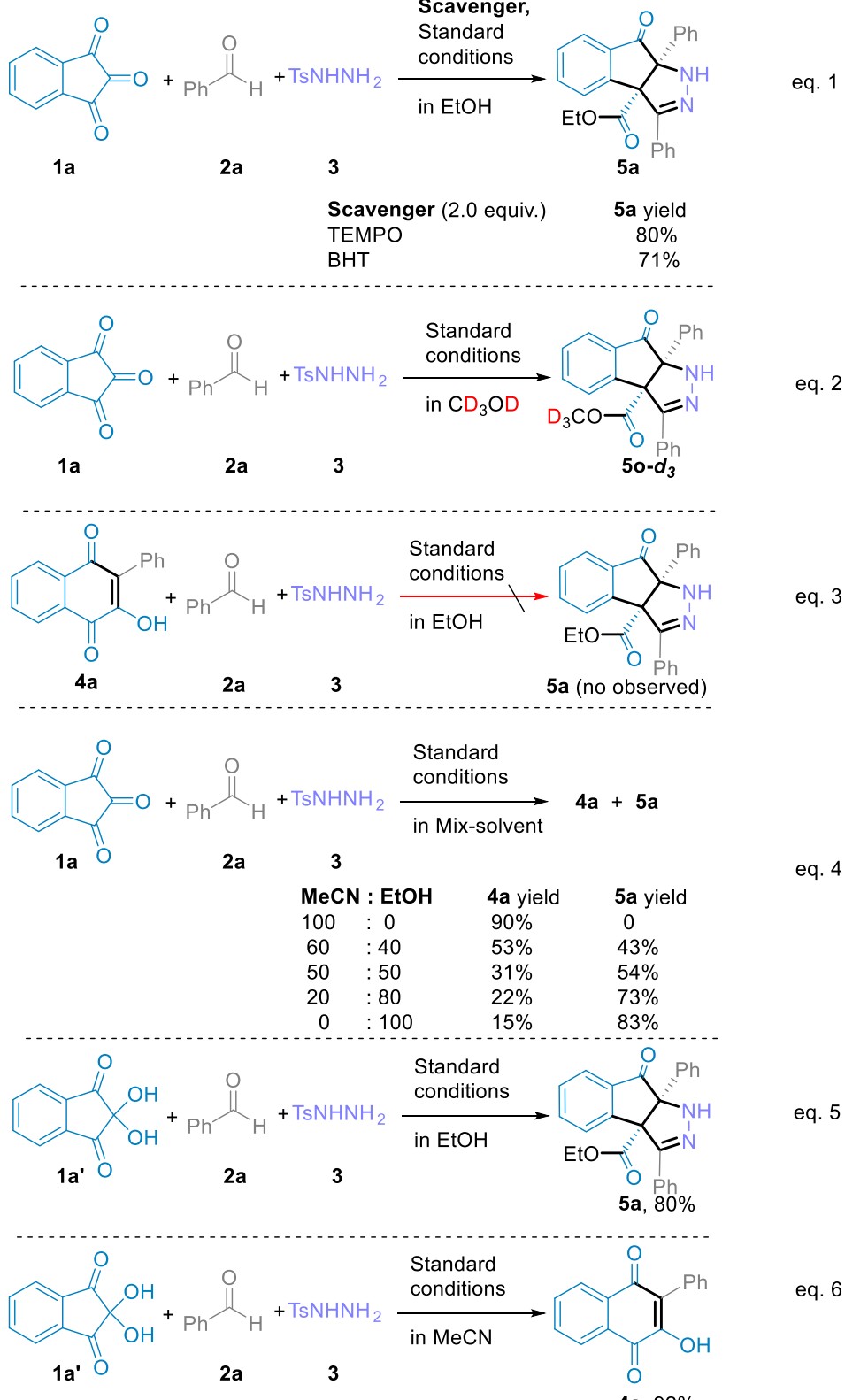

| Scavenger (2.0 equiv.) | 5a yield |
|---|---|
| TEMPO | 80% |
| BHT | 71% |

| MeCN | : | EtOH | 4a yield | 5a yield |
|---|---|---|---|---|
| 100 | : | 0 | 90% | 0 |
| 60 | : | 40 | 53% | 43% |
| 50 | : | 50 | 31% | 54% |
| 20 | : | 80 | 22% | 73% |
| 0 | : | 100 | 15% | 83% |

**Fig. 5 Control experiments.** Free radical capture experiment (eq. 1). Deuterium hydrogen exchange experiment (eq. 2). Standard reaction of intermediate 4 in ethanol solvent (eq. 3). Mixed solvent proportion control experiment (eq. 4). Reaction of ninhydrin hydrate under standard conditions (eqs. 5 and 6).

various control experiments were conducted (Fig. 5). The results indicated that: (1) the addition of TEMPO or BHT did not inhibit the reaction under standard conditions, and no radical-TEMPO/BHT coupling products were detected (Fig. 5, eq. 1); (2) isotope-labeling

experiments in deuterated methanol showed that the ester group of the indenopyrazolone product was formed by bonding the C1 atom of indantrione with deuterated methyl (Fig. 5, eq. 2); (3) using lawsone **4a** and α-aryldiazomethane as the starting material in alcohol solution,

**Fig. 6 Calculated reaction pathways and free energy profiles for the regioselectivity of the ring-expansion process in acetonitrile solvent.** DFT calculations were performed with B3LYP-D3(BJ)-SMD(acetonitrile)/6-311++G(d,p)//B3LYP-D3(BJ)/6-31G(d) (for details see SI, Table S7).

**Fig. 7 Proposed mechanism. a** Reaction formation mechanism of Lawson derivatives. **b** Reaction formation mechanism of indenopyrazole derivatives.

indenopyrazolone product **5a** was not obtained. This shows that the reaction mechanism of the two products proceeded via different paths (Fig. 5, eq. 3); (4) by using the mixed solvent of acetonitrile and ethanol under standard conditions, the yield of the indenopyrazolone product increased gradually upon increasing the proportion of ethanol. On the contrary, the yield of lawsones increased upon increasing the acetonitrile proportion (Fig. 5, eq. 4). This shows that the solvent

had an obvious selectivity for the two different products; (5) Using ninhydrin **1a'** as the starting material, the indenopyrazolone product was also obtained in an alcohol solvent under standard conditions (Fig. 5, eq. 5). The lawsone product was obtained in acetonitrile solvent (Fig. 5, eq. 6).

To better understand the regioselectivity of the ring-expansion process and the cascade reaction mechanism, based on the

**Fig. 8 Calculated reaction pathways and free energy profiles for the regioselectivity of the ring-expansion process in ethanol solvent.** DFT calculations were performed with B3LYP-D3(BJ)-SMD(ethanol)/[other:6-311++G(d,p);Cs: Lanl2DZ] //B3LYP-D3(BJ)/[other: 6-31G (d); Cs: Lanl2DZ] (for details see SI, Table S8).

experimental results, we performed theoretical calculations (see Supplementary Data 8) on the possible reaction pathways of this solvent-controlled regioselective ring-expansion reaction of indantrione and α-aryldiazomethane[42–44]. Initially, aldehyde **2** and *p*-toluenesulfonyl hydrazide **3** condensed in solution to form aryl alkylenediamine **a** (Fig. 6). Then, the aryl alkenylamine removed the benzosulfonyl group to obtain α-aryldiazomethane dipoles **b** and **b'**. Subsequently, indantriones underwent two different ring-expansion reactions in different solvents. In acetonitrile, the ring-expansion reaction of indantrione proceeded via the competitive pathways A and B. In reaction pathway B, the carboanion of the α-aryldiazomethane dipole **b** nucleophile attacked the carbonyl C1 atom of indantrione. Then, intermediate **INT-4** formed through the cycloaddition of transition state TS-3 ($\Delta G = 28.2$ kcal mol$^{-1}$). Next, intermediate **INT-4** underwent intramolecular ring expansion and denitrogenation. After transition state TS-4 ($\Delta G = 30.6$ kcal mol$^{-1}$), intermediate **INT-5** formed, which underwent enol tautomerism to obtain **4a'** (Fig. 6, pathway B). Similarly, in reaction pathway A, α-aryldiazomethane dipole **b** attacked carbonyl C2 of indantrione via transition state TS-1 ($\Delta G = 22.0$ kcal mol$^{-1}$) to form intermediate **INT-2**. Then, the intermediate underwent intramolecular ring expansion and denitrogenation. After transition state TS-2

($\Delta G = 22.1$ kcal mol$^{-1}$), intermediate **INT-3** formed, and finally, tautomerism occurred to form stable product **4a**. Notably, our calculations showed that the energy barrier of the initial step TS-3 ($\Delta G = 22.0$ kcal mol$^{-1}$) of pathway B was higher than TS-1 ($\Delta G = 15.8$ kcal mol$^{-1}$) of pathway A. The energy barrier difference of the two-step transition state was about 6.2 kcal mol$^{-1}$ (Fig. 6). The results indicated that nucleophilic addition between indantrione and α-aryldiazomethane was more favorable at the C2 site than at the C1 site. The reaction was more likely to produce reaction product **4a**. This is consistent with the experimental results. Based on the calculation results, the proposed mechanism is shown in Fig. 7, path a.

In alcohol solution, the alcohol oxygen anion first nucleophilically attacked the carbonyl C1 atom of indantriones to form intermediate **INT-6** (Fig. 8). Then, the intermediate underwent a 1,2-aryl migration rearrangement to obtain the four-membered-ring intermediate, **INT-7** via TS-5 ($\Delta G = 20.1$ kcal mol$^{-1}$). α-Aryldiazomethane **b** was then nucleophilically added to the cyclocarbonyl carbon atom of intermediate **INT-7** to obtain **INT-9** via TS-6 ($\Delta G = 41.2$ kcal mol$^{-1}$). Next, under the catalysis of Cs$_2$HCO$_3$$^+$, **INT-9** underwent transition state TS-7 ($\Delta G = -52.8$ kcal mol$^{-1}$) and TS-8 ($\Delta G = -57.49$ kcal mol$^{-1}$) to undergo

**Fig. 9 Synthetic application.** Multigram-scale experiments and synthetic versatility of lawsones and indenopyrazolones. General conditions: see SI, Supplementary Method 2.

dehydration to form intermediate **INT-11**. Subsequently, α-aryldiazomethane dipole **b** attacked the conjugated double bond of intermediate **INT-11** via TS-9 ($\Delta G = -55.5$ kcal mol$^{-1}$) to obtain **INT-13**. This process was diastereoselective. Finally, intermediate **INT-13** underwent heterocyclic condensation and isomerization to obtain product **5a** (Fig. 8). Based on the calculation results, the proposed mechanism is shown in Fig. 7, path b.

**Synthetic application.** To test the robustness of this solvent-controlled regioselective ring-expansion reaction, a series of multigram-scale experiments were performed (Fig. 9) (see Supplementary Method 1). When 10 mmol of **1a'** and 21 mmol of **2a** in EtOH or MeOH solvent were subjected to the standard reaction conditions, product **5a** or **5o** was isolated in 76% and 81% yield, respectively. Additionally, when 10 mmol of **1a** and 12 mmol of **2a** in MeCN solvent were subjected to the standard reaction conditions, product **4a** was isolated in an 89% yield. The synthetic versatility of lawsones and indenopyrazolones was then explored in an array of derivatizations (see Supplementary Method 2). The carbonyl group of compound **5a** was reduced to a hydroxyl group **6a** under lithium aluminum hydride reduction, but the ester group was unaffected. Compound **6b** was obtained by the selective methoxy stripping of compound **5n** under the action of boron tribromide. Lawsone product **4a** can be used as a reaction synthon to construct a variety of products with diverse structures that can react with phenylalkynes to construct benzannulated bicyclo[3.3.0] octanes compound **6d** with biological activity[45]. Compound **4a** also reacted with diphenylacetylene to obtain alkylidene phthalide **6c**, which is a key intermediate for the synthesis of bioactive natural products[46]. The synthesized bioactive natural product **4a** was directly acetylated to obtain hydroxyacetyl product **6e**. Notably, compound **4w** easily formed carbazoloquinone derivative **6f** under metal catalysis, which is of great interest as a privileged structure for anticancer drug molecules[47].

## Conclusions

In summary, we have described the first solvent-controlled regioselective ring-expansion reaction of indantrione and α-aryldiazomethanes. This reaction preferentially provides the 1,2-carbonyl migration product. In acetonitrile solvent, the reaction products were lawsone derivatives. In alcohol solution, the reaction products were stereoselective indenopyrazolone derivatives. A series of substrates underwent the reaction smoothly, providing highly-functionalized lawsone derivatives and indenopyrazolone derivatives in high yields (up to 95%) and with high levels of diastereoselectivity (up to 95:5 dr). A range of functional groups was also tolerated under the mild reaction conditions. The mechanism control experiment was used to determine the possible reaction mechanism pathways. The synthetic application example also demonstrated the prospect of this method for preparing valuable compounds.

## Methods

**General procedure for preparing lawsones.** A mixture of aldehydes (**2**, 1.2 mmol) and *p*-toluenesulfonyl hydrazide (1.2 mmol) in acetonitrile (20 ml) was stirred at room temperature for 0.5 h until complete consumption of starting materials (monitored by TLC). Then, indantrione (**1**, 1 mmol) and Cs₂CO₃ (2 equiv) were added to the crude product and stirred at 80 °C for 3 h. After the reaction was finished, the solvent was removed under reduced pressure and the residue was purified by silica gel column chromatography (petroleum ether/ethyl acetate 8:1) to afford the desired product **4**. The products were further identified by FTIR spectroscopy, NMR spectroscopy, and HRMS, see Supplementary Data 7.

**General procedure for preparing Indenopyrazoles.** A mixture of aldehydes (**2**, 2.1 mmol) and *p*-toluenesulfonyl hydrazide (2.1 mmol) in alcohol (30 ml) was stirred at room temperature for 0.5–1 h until complete consumption of starting materials (monitored by TLC). Then, indantrione (**1**, 1 mmol) and Cs₂CO₃ (3 equiv) were added to the crude product and stirred at 80 °C for 10 h. After the reaction was finished, the solvent was removed under reduced pressure and the residue was purified by silica gel column chromatography (petroleum ether/ethyl acetate 25:1) to afford the desired product **5**. The products were further identified by FTIR spectroscopy, NMR spectroscopy, and HRMS (see Supplementary Data 7).

## Data availability

The authors declare that the data supporting the findings of this study are available within the article and the Supplementary Information as well as from the authors upon reasonable request. The X-ray crystallographic coordinates for structures 4h, 4ae, 5a, 5f, 5q, and 5r, reported in this study have been deposited at the Cambridge Crystallographic Data Centre (CCDC), under CCDC 2149716 (4h, Supplementary Data 1), 2149715 (4ae, Supplementary Data 2), 2149717 (5a, Supplementary Data 3), 2149718 (5f, Supplementary Data 4), 2149719 (5q, Supplementary Data 5) and 2149720 (5r, Supplementary Data 6), respectively. These data can be obtained free of charge from The Cambridge Crystallographic Data Centre via www.ccdc.cam.ac.uk/data_request/cif. The compound characterizations are available in Supplementary Data 7. DFT calculations are available in Supplementary Data 8. The computed output file is in the open database (https://zenodo.org/) link (DOI:10.5281/zenodo.7296393).

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

## Acknowledgements

The authors gratefully acknowledge for financial support of the Program for the NSFC (Nos. 22267021, 22067020), Yunnan Fundamental Research Projects (202101AS070034, 202001BB050009), and the Program for Excellent Young Talents, Yunnan University.

## Author contributions

Y.J. conceived and directed the project. BW.H., PY.J., L.J., and X.Y. performed the experiments. WX.Y. performed the theoretical calculations. J.L., YC.J., and Y.J. analyzed the results, and Y.J. wrote the manuscript.

## Competing interests

The authors declare no competing interests.
