## [Peer Review File · Communications Chemistry]

Reviewers' comments:

Reviewer #1 (Remarks to the Author):

The manuscript by Hu et al. is an interesting paper reporting ring-expansion reactions of indantrione, leading to either lawsones or indenopyrazoles. The paper is reasonably well written, and reports a thorough scope and mechanism exploration, using both experimental and computational methods. Both compound classes have interesting biological activities. Therefore, the paper seems of sufficient general interest and quality to potentially warrant a publication in Nature Communications Chemistry. However, there are various issues that need to be addressed first.

Specific things to be addressed below:

- 1) "Since scaffold generation is pivotal during the early stage of drug discovery, issues like chemical efficiency, and easy diversification are crucial in process development"
Process development is ambiguous in this context - is it meant as large-scale drug manufacturing, or as reaction development?
- 2) "classical homology method"  "classical homologation method"
- 3) "for generate structurally"  "for generation of structurally"
- 4) Please expand Scheme 1 to include specific illustrations of previous approaches (e.g. from refs 4, 5, 6, 7). This will allow readers to better gauge the novelty of the current work.
- 5) " imbedded"  "embedded"
- 6) "that aren't readily available"  "that are not readily available"
- 7) "A small amount of selective by-product 4a was generated in 15% yield. " This is confusing - how can a by-product be selective?
- 8) "two different chemoselective products" - I suggest leaving out 'chemoselective'.
- 9) "Table 1 Optimized reaction conditions." - I do not think that this caption is appropriate, perhaps 'reaction optimization'?
- 10) "To verity the relative configurations" -- > "To verify the relative configurations"
- 11) "facilely" - spelling
- 12) " labeling" - spelling
- 13) Page 8, line 120: " using lawsone product 4a as the starting material, the reaction proceeded under the 121 standard conditions in an alcohol solution." The phrasing here suggests that the reaction worked, but the scheme clearly shows that this reaction did not produce 5a. Please consider rephrasing.
- 14) Page 9, line 125: "the yield of the indenopyrazolone product increased gradually upon increasing the

proportion of acetonitrile. " Scheme 4 shows the complete opposite - that the yield of the indenopyrazolone increased with the proportion of ethanol.

15) Page 9, line 127: "On the contrary, the yield of lawsones increased upon 127 increasing the methanol proportion" Again, the complete opposite is shown in Scheme 4, eq 4. Also, the scheme shows ethanol was used, but the text mentions methanol - which was actually used?

16) Energies are quoted in text and schemes to 2 d.p. precision - this implies much higher accuracy than DFT can deliver. I believe the energies should be quoted to 1 d.p. precision.

17) Scheme 5 - the highest barrier in DFT calculated pathway is 22.1 kcal/mol, which suggests the reaction can proceed at room temperature or maybe slightly above, but the reaction is run at 80C. Please speculate on the reasons behind this discrepancy.

18) Scheme 4 - 'Standanrd' spelling in all 6 instances

19) At the start of the computational section, please briefly mention the main computational methods used (functional, basis set, solvent model, software).

20) Scheme 6: structure 5a is cropped in the PDF.

21) Scheme 6: in structures INT-6 and TS-5 the EtO group is on the bottom of the structures, but in INT-7, INT-8, TS-6, INT-9, TS-7 the EtO group is on top. Please flip either the structures in the beginning or the end to keep it consistent and clear.

22) "INT-7 (detected by HRMS)" - could this MS peak also be protonated INT-6? It is not clear how the two could be distinguished simply by HRMS.

23) Scheme 7, path b - again, it should be noted how can the rearranged cyclobutane intermediate be distinguished from protonated preceding intermediate - I think both should give $[M+H]^+$: 207.0658.

24) "Scheme 7 Proposed machenism."  spelling

Supporting Information:

25) Integration in all of the ^1H NMR copies should be adjusted. The integration curves must have flat ends, indicating that the integration bounds have been set properly - this is not the case in most copies provided by the authors. Also, it makes no mathematical sense integrating parts of an overlapping multiplet - overlapping peaks should be integrated as one large multiplet.

26) Main text needs a comment on the purity of the p-NO₂ compound in SI page 74. This sample should be repurified, an updated NMR collected and updated yield after repurification reported in text. Also, NMR copy should show all of the NMR range, including the 0 - 2ppm range, just like most other NMR copies in this manuscript.

27) Page 80 of the SI - the integration of the methyl peak in the ^1H NMR is particularly bad. It is obvious only a tiny sliver of the actual peak has been integrated. Please reprocess the NMR.

28) Page 84 - CN- compound needs repurification and yield adjustment, NMR clearly shows impure sample.

29) Page 104 - compound needs repurification and yield adjustment, NMR clearly shows impure

sample.

30) Gaussian reference is incomplete - their website provides the preferred format for referencing the software.

31) I strongly encourage the authors to upload their computational output files as a dataset to a site like <https://zenodo.org/>, or a similar open data repository and include the DOI link the dataset in the revised manuscript. Alternatively, the output files could be included as a data archive and attached as part of the SI. Sharing the output files makes the work more accessible and easier to reproduce and build upon, thus increasing its impact. The Cartesian coordinates are difficult to use, contains much more limited data, and presents an unnecessary barrier to reproduction of the results.

The paper by Hu et al. provides an interesting synthetic and mechanistic study of lawsones and indenopyrazole derivatives via ring-expansion reactions. The paper has some issues with spelling, data interpretation and reporting and analytical data. However, after these issues are addressed, I believe the paper would be suitable for publication in Nature Communications Chemistry.

Reviewer #2 (Remarks to the Author):

The authors describe herein the switchable synthesis of lawsones and indenopyrazoles via regioselective ring-expansion of indantrione with aldehydes using an umpolung strategy. The chemistry developed in this manuscript is intriguing and features profound utility potential for assembling such skeletons, which should of interest to the readers of Communications Chemistry. However, the computational mechanistic studies for compound 5a seem to be unreasonable as the highest energy barrier is up to 57.49 kcal/mol (from INT-6 to TS-7), which is inconceivable for a reaction conducted at 80 oC. In this view, I recommend that the authors should revise this issue and propose a more reasonable reaction path before the acceptance.

Other issues should also be addressed:

- 1) p-Methylbenzenesulfonohydrazide was used for the in situ formation of α -aryldiazomethane dipoles, have the authors tried hydrazine hydrate as its replacement? Hydrazine hydrate should be more green and practical compared with p-methylbenzenesulfonohydrazide.
- 2) Scheme 3, alkyl-substituted aldehydes should also be tested to probe the generality or limitation of this protocol.
- 3) Scheme 6, the free energy of INT-10 and INT-12 was wrong-labeled.
- 4) For the supporting information file, the spectra for many compounds contain obvious impurities, e.g. 4e, 4f, 4h, 4k, 4ae, 4ai, 4ak etc, which should be further re-purified.

Reviewer #3 (Remarks to the Author):

Dear Editorial Team of Communications Chemistry,

In attention to manuscript ID COMMSCHEM-22-0287-T entitled " Switchable Synthesis of Natural-Product-Like Lawsones and Indenopyrazoles through Regioselective Ring-Expansion of Indantrione", the authors present the synthesis of lawsones and indenopyrazole derivatives in aprotic (acetonitrile) and protic (alcohol) solvents through a new metal-free, ring-expansion reaction of indantrione with diazomethanes, generated in situ from the N-tosylhydrazones.

The paper is complete. The authors explored all reaction pathways to demonstrate the mechanism of

formation of the two products through an experimental and in silico study. Finally, they also demonstrated the synthetic versatility of lausone and indenopyrazoles. However, the synthesis of lausone derivatives of indantrione is not new and deserves mention. Schanck et al. reported a very similar reaction in *Helv. Chim. Acta* 2001, 84 (7), 2071-2088 ([https://doi.org/10.1002/1522-2675\(20010711\)84:7<2071::AID-HLCA2071<3.0.CO;2-D](https://doi.org/10.1002/1522-2675(20010711)84:7<2071::AID-HLCA2071<3.0.CO;2-D)).

It is noteworthy that this opinion was given without consideration of errors in grammar and spelling or standard formatting for *Communications Chemistry*.

Finally, I recommend publishing the article on *Communications Chemistry*.

雲南大學

YUNNAN UNIVERSITY

Kunming, Yunnan, The People's Republic of China, 650091

Nov. 08, 2022

Response to Reviewers

(Manuscript #: COMMSCHEM-22-0287-T)

Dear Reviewers,

Thank you for your nice consideration of our manuscript entitled "Switchable Synthesis of Natural-Product-Like Lawsones and Indenopyrazoles through Regioselective Ring-Expansion of Indantrione". Those comments are all valuable and very helpful for revising and improving our manuscript. We have read the comments carefully and have made corrections which we hope meet with approval. Revised portions are marked in **Yellow** in the manuscript and Supporting Information (SI) file. The responses to your comments are as follows.

Reviewer 1:

Q-1) "Since scaffold generation is pivotal during the early stage of drug discovery, issues like chemical efficiency, and easy diversification are crucial in process development"

Process development is ambiguous in this context - is it meant as large-scale drug manufacturing, or as reaction development?

Response:

Thanks for reviewer's comments. The "Process development" meant as reaction development. We have modified it in manuscript.

Q-2) "classical homology method"  "classical homologation method"

Response:

We have revised this error in manuscript.

雲南大學

YUNNAN UNIVERSITY

Kunming, Yunnan, The People's Republic of China, 650091

Q-3) "for generate structurally"  "for generation of structurally"

Response:

We have revised this error in manuscript.

Q-4) Please expand Scheme 1 to include specific illustrations of previous approaches (e.g. from refs 4, 5, 6, 7). This will allow readers to better gauge the novelty of the current work.

Response:

We have expanded the Scheme 1 to include two specific examples.

Q-5) " imbedded"  "embedded"

Response:

We have corrected this error in manuscript.

Q-6) "that aren't readily available"  "that are not readily available"

Response:

We have corrected this error in manuscript.

Q-7) "A small amount of selective by-product 4a was generated in 15% yield. " This is confusing - how can a by-product be selective?

Response:

We have corrected this misdescription in manuscript.

Q-8) "two different chemoselective products" - I suggest leaving out 'chemoselective'.

Response:

Thanks for your suggestion, we have deleted 'chemoselective' in this sentence.

雲南大學

YUNNAN UNIVERSITY

Kunming, Yunnan, The People's Republic of China, 650091

Q-9) "Table 1 Optimized reaction conditions." - I do not think that this caption is appropriate, perhaps 'reaction optimization'?

Response:

Thanks for your suggestion, we have changed this caption to 'Reaction optimization'.

Q-10) "To verity the relative configurations" -- > "To verify the relative configurations"

Response:

We have corrected this error in manuscript.

Q-11) "facilely" - spelling

Response:

We have changed this word as "smoothly".

Q-12) " labeling" - spelling

Response:

We have changed this word as "isotope-labeling".

Q-13) Page 8, line 120: " using lawsone product 4a as the starting material, the reaction proceeded under the standard conditions in an alcohol solution." The phrasing here suggests that the reaction worked, but the scheme clearly shows that this reaction did not produce 5a. Please consider rephrasing.

Response:

We have corrected this misdescription in manuscript.

Q-14) Page 9, line 125: "the yield of the indenopyrazolone product increased gradually upon increasing the proportion of acetonitrile. " Scheme 4 shows the

雲南大學

YUNNAN UNIVERSITY

Kunming, Yunnan, The People's Republic of China, 650091

complete opposite - that the yield of the indenopyrazolone increased with the proportion of ethanol.

Response:

We have matched the text description to the experiment results shown in Scheme 4, Eq. 4.

Q-15) Page 9, line 127: "On the contrary, the yield of lawsones increased upon increasing the methanol proportion" Again, the complete opposite is shown in Scheme 4, eq 4. Also, the scheme shows ethanol was used, but the text mentions methanol - which was actually used?

Response:

We have matched the text description to the experiment results shown in Scheme 4. Mixed solvent of acetonitrile and ethanol was used to the experiment, and we have corrected the relevant misdescription in this statement.

Q-16) Energies are quoted in text and schemes to 2 d.p. precision - this implies much higher accuracy than DFT can deliver. I believe the energies should be quoted to 1 d.p. precision.

Response:

The accuracy of energy involved in this article has been revised to 1 d.p. precision.

17) Scheme 5 - the highest barrier in DFT calculated pathway is 22.1 kcal/mol, which suggests the reaction can proceed at room temperature or maybe slightly above, but the reaction is run at 80C. Please speculate on the reasons behind this discrepancy.

Response:

At room temperature, the highest barrier in DFT calculation is 20.87 kcal/mol, However, no corresponding product was obtained in the experiment, The specific

雲南大學

YUNNAN UNIVERSITY

Kunming, Yunnan, The People's Republic of China, 650091

reasons need to be further studied in the future.

Q-18) Scheme 4 - 'Standanrd' spelling in all 6 instances

Response:

We have corrected this error in manuscript.

Q-19) At the start of the computational section, please briefly mention the main computational methods used (functional, basis set, solvent model, software).

Response:

We have added basis set and solvent model in the calculation at the appropriate positions in the article (Scheme 5 and 6)

Q-20) Scheme 6: structure 5a is cropped in the PDF.

Response:

Scheme 6 has been adjusted so that structure **5a** can be fully displayed.

Q-21) Scheme 6: in structures INT-6 and TS-5 the EtO group is on the bottom of the structures, but in INT-7, INT-8, TS-6, INT-9, TS-7 the EtO group is on top. Please flip either the structures in the beginning or the end to keep it consistent and clear.

Response:

We adjusted Structures INT-6,TS-5,INT-7,INT-8,TS-6 in Scheme 6 to keep the EtO group at the bottom.

Q-22) "INT-7 (detected by HRMS)" - could this MS peak also be protonated INT-6? It is not clear how the two could be distinguished simply by HRMS.

Response:

We have deleted the corresponding MS data and give a new mechanism in the

雲南大學

YUNNAN UNIVERSITY

Kunming, Yunnan, The People's Republic of China, 650091

manuscript.

Q-23) Scheme 7, path b - again, it should be noted how can the rearranged cyclobutane intermediate be distinguished from protonated preceding intermediate - I think both should give $[M+H]^+$: 207.0658.

Response:

Thanks for your suggestion, we have deleted the corresponding MS data and give a new mechanism in the manuscript.

Q-24) "Scheme 7 Proposed machenism."  spelling

Response:

We have corrected this error in manuscript.

Q-25) Integration in all of the 1H NMR copies should be adjusted. The integration curves must have flat ends, indicating that the integration bounds have been set properly - this is not the case in most copies provided by the authors. Also, it makes no mathematical sense integrating parts of an overlapping multiplet - overlapping peaks should be integrated as one large multiplet.

Response:

Thanks for your suggestion, we have rechecked all of the 1H NMR data and adjusted almost all of the corresponding integration.

Q-26) Main text needs a comment on the purity of the p-NO₂ compound in SI page 74. This sample should be repurified, an updated NMR collected and updated yield after repurification reported in text. Also, NMR copy should show all of the NMR range, including the 0 - 2ppm range, just like most other NMR copies in this manuscript.

雲南大學

YUNNAN UNIVERSITY

Kunming, Yunnan, The People's Republic of China, 650091

Response:

We have re-purified compound **4k** and updated yield and NMR spectra in SI.

Q-27) Page 80 of the SI - the integration of the methyl peak in the ¹H NMR is particularly bad. It is obvious only a tiny sliver of the actual peak has been integrated. Please reprocess the NMR.

Response:

We have adjusted the integration of **4p** in SI.

Q-28) Page 84 - CN- compound needs repurification and yield adjustment, NMR clearly shows impure sample.

Response:

We have re-purified compound **4s** and updated yield and NMR spectra in SI.

Q-29) Page 104 - compound needs repurification and yield adjustment, NMR clearly shows impure sample.

Response:

We have repurified compound **4ak** and updated yield and NMR spectra in SI.

Q-30) Gaussian reference is incomplete - their website provides the preferred format for referencing the software.

Response:

Gaussian reference format has been supplemented completely.

Q-31) I strongly encourage the authors to upload their computational output files as a dataset to a site like <https://zenodo.org/>, or a similar open data repository and include the DOI link the dataset in the revised manuscript. Alternatively, the output files could

雲南大學

YUNNAN UNIVERSITY

Kunming, Yunnan, The People's Republic of China, 650091

be included as a data archive and attached as part of the SI. Sharing the output files makes the work more accessible and easier to reproduce and build upon, thus increasing its impact. The Cartesian coordinates are difficult to use, contains much more limited data, and presents an unnecessary barrier to reproduction of the results.

Response:

The computational output files have been uploaded to the open database <https://zenodo.org/>, and the DOI link the dataset is inserted into the calculation details in the SI

Reviewer 2:

Q-1) p-Methylbenzenesulfonohydrazide was used for the in situ formation of α -aryldiazomethane dipoles, have the authors tried hydrazine hydrate as its replacement? Hydrazine hydrate should be more green and practical compared with p-methylbenzenesulfonohydrazide.

Response:

Thanks for reviewer's comments. We have tried replacing p-toluenesulfonyl hydrazide with hydrazine hydrate, but unfortunately, we did not get satisfying results.

Q-2) Scheme 3, alkyl-substituted aldehydes should also be tested to probe the generality or limitation of this protocol.

Response:

Thanks for your suggestion, we screened prenylaldehyde, butyraldehyde and 2-ethylcaproaldehyde in this reaction, unfortunately, alkyl-substituted aldehydes were found to be incompatible with this transformation and did not yield any product, even after stirring for more than 24 h. We have made comments in the manuscript.

Q-3) Scheme 6, the free energy of INT-10 and INT-12 was wrong-labeled.

雲南大學

YUNNAN UNIVERSITY

Kunming, Yunnan, The People's Republic of China, 650091

Response:

We have corrected this error in manuscript.

Q-4) For the supporting information file, the spectra for many compounds contain obvious impurities, e.g. 4e, 4f, 4h, 4k, 4ae, 4ai, 4ak etc, which should be further re-purified.

Response:

We have re-purified compound 4e, 4f, 4h, 4k, 4s, 4ae, 4ai, 4ak and updated NMR spectra in SI.

Reviewer 3:

Q-1) The paper is complete. The authors explored all reaction pathways to demonstrate the mechanism of formation of the two products through an experimental and in silico study. Finally, they also demonstrated the synthetic versatility of lausone and indenopyrazoles. However, the synthesis of lausone derivatives of indantrione is not new and deserves mention. Schanck et al. reported a very similar reaction in *Helv. Chim. Acta* 2001, 84 (7), 2071-2088 ([https://doi.org/10.1002/1522-2675\(20010711\)84:7<2071::AID-HLCA2071<3.0.CO;2-D](https://doi.org/10.1002/1522-2675(20010711)84:7<2071::AID-HLCA2071<3.0.CO;2-D)).

It is noteworthy that this opinion was given without consideration of errors in grammar and spelling or standard formatting for *Communications Chemistry*.

Finally, I recommend publishing the article on *Communications Chemistry*.

Response:

Thank the reviewer for your positive comments on our research work. We added the reference mentioned by the reviewer as "Ref. 15d".

雲南大學

YUNNAN UNIVERSITY

Kunming, Yunnan, The People's Republic of China, 650091

Once again, thank you very much for your comments and suggestions, and look forward to your reply.

Yours sincerely,

Prof. Dr. Yi Jin

Key Laboratory of Medicinal Chemistry for Natural Resource (Yunnan University)

Ministry of Education; School of Chemical Science and Technology

Yunnan University, Kunming, 650091, P. R. China.

Tel & Fax: +86 871 65031633

E-mail: jinyi@ynu.edu.cn

REVIEWERS' COMMENTS:

Reviewer #1 (Remarks to the Author):

The manuscript by Hu et al. is an interesting paper reporting ring-expansion reactions of indantrione, leading to either lawsones or indenopyrazoles . The paper is reasonably well written, and reports a thorough scope and mechanism exploration, using both experimental and computational methods.

The authors seem to have addressed all of the referees points. In particular, I thank the authors for the repurification and reprocessing of the NMRs and for the sharing of the computational data.

In light of this, I would recommend the paper for publication in Nature Communications Chemistry in its current form.